# Morbidity and unplanned healthcare encounters after hospital discharge among young children in Dar es Salaam, Tanzania and Monrovia, Liberia

Rodrick Kisenge,[1] Readon C Ideh,[2] Julia Kamara,[2] Ye-Jeung G Coleman-Nekar,[2] Abraham Samma,[1] Evance Godfrey,[1] Hussein K Manji,[3,4] Christopher R Sudfeld,[5] Adrianna Westbrook,[6] Michelle Niescierenko,[7,8] Claudia R Morris,[9,10] Cynthia G Whitney,[11] Robert F Breiman,[12,13] Christopher P Duggan,[5,14] Karim P Manji,[1] Chris A Rees [ORCID] [9,10]

CPD, KPM and CAR are joint senior authors.

**Correspondence to**
Dr Rodrick Kisenge; saroriki@yahoo.com

## ABSTRACT

**Background** Researchers and healthcare providers have paid little attention to morbidity and unplanned healthcare encounters for children following hospital discharge in low- and middle-income countries. Our objective was to compare symptoms and unplanned healthcare encounters among children aged <5 years who survived with those who died within 60 days of hospital discharge through follow-up phone calls.

**Methods** We conducted a secondary analysis of a prospective observational cohort of children aged <5 years discharged from neonatal and paediatric wards of two national referral hospitals in Dar es Salaam, Tanzania and Monrovia, Liberia. Caregivers of enrolled participants received phone calls 7, 14, 30, 45, and 60 days after hospital discharge to record symptoms, unplanned healthcare encounters, and vital status. We used logistic regression to determine the association between reported symptoms and unplanned healthcare encounters with 60-day post-discharge mortality.

**Results** A total of 4243 participants were enrolled and had 60-day vital status available; 138 (3.3%) died. For every additional symptom ever reported following discharge, there was a 35% greater likelihood of post-discharge mortality (adjusted odds ratio [aOR] 1.35, 95% confidence interval [CI] 1.10 to 1.66; p=0.004). The greatest survival difference was noted for children who had difficulty breathing (2.1% among those who survived vs 36.0% among those who died, p<0.001). Caregivers who took their child home from the hospital against medical advice during the initial hospitalisation had over eight times greater odds of post-discharge mortality (aOR 8.06, 95% CI 3.87 to 16.3; p<0.001) and those who were readmitted to a hospital had 3.42 greater odds (95% CI 1.55 to 8.47; p=0.004) of post-discharge mortality than those who did not seek care when adjusting for site, sociodemographic factors, and clinical variables.

**Conclusion** Surveillance for symptoms and repeated admissions following hospital discharge by healthcare providers is crucial to identify children at risk for post-discharge mortality.

### WHAT IS ALREADY KNOWN ON THIS TOPIC

⇒ There is growing recognition that there is substantial mortality following hospital discharge among young children in sub-Saharan Africa.
⇒ Prior work to identify young children at risk for post-discharge mortality has focused on factors present during the index hospital admission but has not explored the role symptoms or healthcare seeking following hospital discharge may have in post-discharge mortality.

### WHAT THIS STUDY ADDS

⇒ For every additional symptom ever reported among young children following hospital discharge, there was a 35% greater likelihood of post-discharge mortality.
⇒ Young children whose caregivers took them home against medical advice, those who had symptoms and did not seek additional care following hospital discharge, and those who were readmitted had the greatest odds of post-discharge mortality.

### HOW THIS STUDY MIGHT AFFECT RESEARCH, PRACTICE OR POLICY

⇒ Surveillance by healthcare providers following hospital discharge focused on new or persistent symptoms may better identify young children at risk for post-discharge mortality.
⇒ Follow-up for children post-discharge should actively be undertaken for the children whose caregivers left the hospital against medical advice.

## INTRODUCTION

There is emerging evidence that the months following a hospitalisation for an illness represent a vulnerable time in the lives of young children in low- and middle-income countries including those in sub-Saharan Africa. As many as 1%–18% of young children die within 6 months of hospital discharge in sub-Saharan

Africa.[1 2] Prior studies have focused on the identification of risk factors to better identify children during hospitalisation who may be at risk for post-discharge mortality in sub-Saharan Africa.[3–7] Although risk stratification at the time of hospital discharge is immensely important to reduce post-discharge mortality, little attention has been paid to morbidity or unplanned healthcare encounters among children in the post-discharge period.

Results from prior studies suggest that delayed healthcare seeking, often due to long distances to healthcare facilities, lack of transportation, or lack of financial resources to pay for health services, may contribute to high rates of mortality in children in sub-Saharan Africa.[8–12] For example, more children die at home in settings in which patients have to pay for healthcare services than in settings in which they do not have to pay.[11] However, such studies have not focused on healthcare seeking in the vulnerable time following hospital discharge.

Elucidating morbidity and unplanned healthcare encounters among young children who die following hospital discharge has the potential to better identify at-risk children. Moreover, the identification of morbidity following hospital discharge may present an opportunity for interventions to reduce post-discharge mortality among young children in sub-Saharan Africa. To define post-discharge events that could be utilised to trigger interventions to prevent mortality, our objective was to compare morbidity and unplanned healthcare encounters among children aged <5 years who survived within 60 days after hospital discharge with those who died at two sites in sub-Saharan Africa.

## METHODS
### Study design
We conducted a planned secondary analysis of a prospective observational cohort study in which neonates, infants, and young children aged <5 years were enrolled at the time of hospital discharge and were followed up to 60 days through telephone calls made to caregivers (i.e., the individual who accompanied the child during the hospital admission and identified as a primary caregiver). The study protocol and rationale have been previously published.[13] The ethical review boards of the Tanzania National Institute of Medical Research, the Muhimbili University of Health and Allied Sciences Research and Ethics Committee, the John F. Kennedy Medical Centre, Boston Children's Hospital, and Emory University reviewed and approved this study.

### Patient and public involvement statement
The development of the study question was informed by emerging recognition of the burden of post-discharge mortality among young children in sub-Saharan Africa. Patients and caregivers were not involved in the design, recruitment, or conduct of the study, nor were they advisers in this study. Results of this study will be made publicly available through publication.

### Study setting
This study was conducted at Muhimbili National Hospital (MNH) in Dar es Salaam, Tanzania and John F. Kennedy Medical Center (JFKMC) in Monrovia, Liberia from October 2019 to January 2022. MNH is Tanzania's largest referral hospital, is located in an urban area, and has a catchment area of >6 million people (approximately 840 000 children aged <5 years). Similarly, JFKMC is Liberia's largest referral hospital, is also located in an urban area, and serves a catchment area of >1.2 million people (approximately 200 000 children aged <5 years). Both hospitals are supported by the Ministry of Health of each country and serve as major training sites for medical students and residents in paediatrics. Both hospitals provide comprehensive clinical care for ill and injured patients and have surgical services available. These two hospitals were selected because of long-standing collaborative relationships among the investigators, their locations being in similar settings, both serving as national referral hospitals, and having similar numbers of monthly discharges from the neonatal and paediatric wards. Additionally, we aimed to include hospitals in diverse settings in sub-Saharan Africa to overcome inherent limitations in single-centre, or single-region, studies.

### Study population and inclusion and exclusion criteria
Young children aged <5 years were enrolled near the time of hospital discharge (i.e., ≤24 hours before anticipated discharge). Participants aged 0–28 days at the time of discharge were considered neonates and those 29 days to 59 months of age were considered infants and children in this study. Participants were enrolled regardless of the reason for hospitalisation inasmuch as caregivers provided consent to enrolment. Participants who died during the index hospitalisation, those whose caregivers did not have access to a telephone for follow-up calls, and those whose caregivers did not consent were excluded. We aimed to enrol approximately 1000 neonates at each site (n=2000 total) and 1000 infants and children at each site (n=2000 total). The sample size was determined based on estimated post-discharge mortality rates of 5% to develop risk assessment tools including at least five variables to identify (a) neonates and (b) infants and children who were at risk for post-discharge mortality, as described previously.[13]

### Study procedures
We consecutively enrolled participants from the neonatal ward (for neonates) and the paediatric ward (for infants and children) at each site. Caregivers of potentially eligible participants were approached by research staff who described the study procedures in Kiswahili in Tanzania and English in Liberia including telephone follow-up for 60 days after hospital discharge. Prior to any data collection, caregivers who were accompanying participants in the hospital were approached by research staff and asked to provide informed written (in Dar es Salaam) or oral consent (in Monrovia) for the collection

of sociodemographic data, hospital data, and to follow-up calls after hospital discharge. Oral consent was obtained in Monrovia because of cultural preference and low rates of caregiver literacy.

Research staff reviewed the medical record of enrolled participants to extract clinical variables and conducted brief interviews with caregivers to obtain sociodemographic data not recorded in the medical record. Research staff recorded all data in password-protected, standardised, electronic case report forms in the software SQL (Microsoft, Seattle, WA) in Tanzania and KoboToolbox (KoboToolbox, Cambridge, MA) in Liberia.

Research staff made telephone calls to enrolled caregivers 7, 14, 30, 45, and 60 days from the time of hospital discharge. During these phone calls, research staff first confirmed that the respondent was the caregiver who had consented to enrolment then inquired about common symptoms and asked about any unplanned healthcare encounters (e.g., clinic visits for illnesses, unplanned hospital admissions, etc.) during each follow-up call using a standard case report form. The primary outcome was caregiver reported all-cause, 60-day post-discharge mortality. If a participant had died, the same questions about preceding symptoms and healthcare encounters were asked during follow-up calls. If caregivers did not respond to telephone calls, research staff made two additional telephone calls followed by a text message and a home visit if there was no response.

## Statistical analyses

Descriptive statistics for participant demographics were calculated. Medians and IQRs were determined for continuous variables and proportions were calculated for categorical variables. Continuous variables were compared using the Wilcoxon rank-sum test. Frequencies of available reported symptoms and unplanned healthcare encounters among children who survived within 60 days of hospital discharge and those who died were compared using $\chi^2$ or Fisher's exact testing. Overall survival between those with symptoms who sought care and those with symptoms who did not seek care at each time point of interest were evaluated using Kaplan-Meier analysis. Greater rates of post-discharge mortality among participants whose caregivers left the hospital against medical advice during the index hospitalisation were noted. Hence, subgroup analyses of this population were conducted to determine morbidity and unplanned healthcare encounters following discharge. Unadjusted and adjusted logistic regression analyses were conducted in order to determine the association between unplanned healthcare encounters and survival. Predictors with an unadjusted p value<0.2 were included

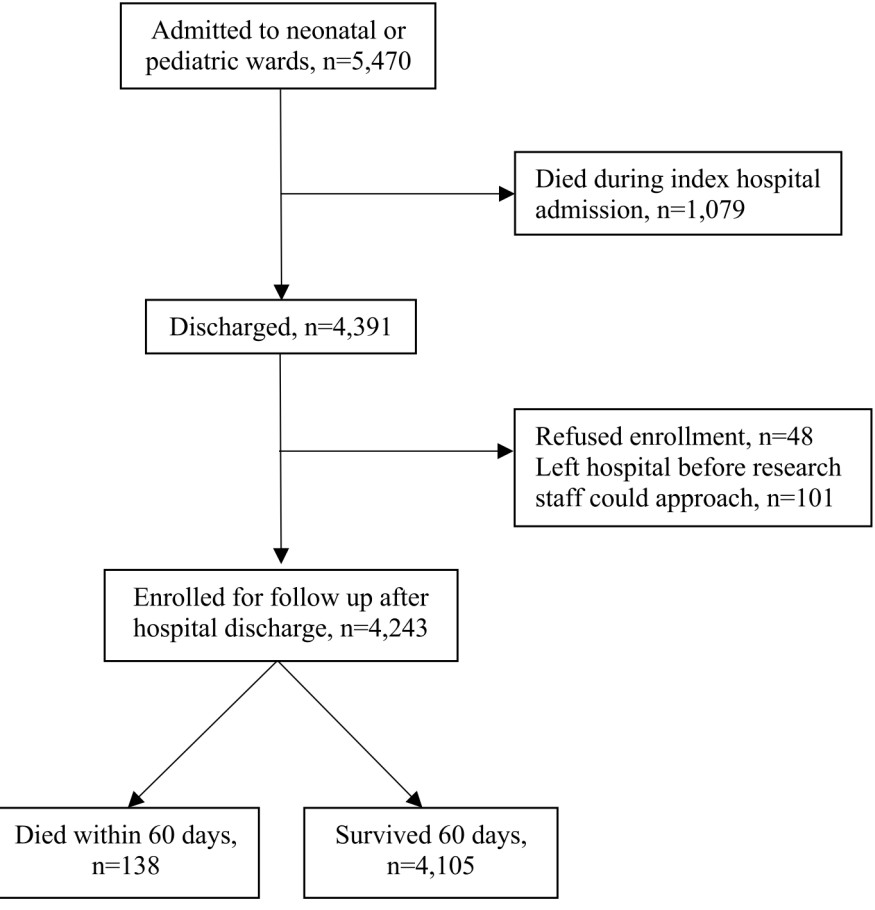

**Figure 1** Flow diagram for included participants.

**Table 1** Demographic characteristics of young children discharged from referral hospitals in Dar es Salaam, Tanzania and Monrovia, Liberia by 60-day survival outcome

| Characteristics | Survived 60 days after discharge, n=4105, n (%) | Died within 60 days after discharge, n=138, n (%) |
|---|---|---|
| Site (n=4243) | | |
| Liberia | 2174 (53) | 75 (54) |
| Tanzania | 1931 (47) | 63 (46) |
| Sex (n=4232) | | |
| Female | 1803 (44) | 56 (42) |
| Male | 2295 (56) | 78 (58) |
| Age at discharge, months (median, IQR) (n=4243) | 1 (0, 9) | 1 (0, 7) |
| Number of discharge diagnoses | | |
| 1 only | 1428 (35) | 41 (30) |
| 2 only | 1427 (35) | 46 (33) |
| 3 or more | 1250 (30) | 51 (37) |
| Infectious diagnosis during hospital admission | 2519 (61) | 83 (60) |
| Disposition from hospital (n=4241) | | |
| Discharge | 3977 (97) | 116 (85) |
| Against medical advice | 125 (3.0) | 20 (15) |
| Transfer | 3 (<0.1) | 0 (0) |
| Time to nearest health facility, hours (median, IQR) | 0.25 (0.17, 0.50) | 0.33 (0.25, 0.50) |
| Type of health facility nearest home (n=3837) | | |
| Pharmacy | 2176 (59) | 73 (61) |
| Clinic | 1124 (30) | 32 (27) |
| Hospital | 386 (10) | 14 (12) |
| Other | 32 (0.9) | 0 (0) |
| Caregiver age in years (median, IQR) (n=3867) | 28 (23, 33) | 28 (23, 34) |
| Caregiver education level (n=3892) | | |
| Completed college/university or vocational/technic* | 486 (13) | 9 (7.8) |
| Completed/some primary school | 944 (25) | 26 (23) |
| Completed/some secondary school | 2017 (54) | 65 (57) |
| No formal schooling | 286 (7.6) | 14 (12) |
| Other | 33 (0.9) | 1 (0.9) |

*Vocational/technic denotes training for specific jobs (e.g., electrician).

in the multivariable model. All analyses were performed in R V.4.1.3 (R Foundation for Statistical Computing, Vienna, Austria) and SAS V.9.4 (SAS Institute, Cary, NC).

## RESULTS

There were 5470 neonates, infants, and young children admitted to the two included hospitals during the study period, 19.7% (n=1079) of which died during their index hospital admission. Among the 4391 discharged patients, 4243 (96.7%) enrolled and had 60-day outcomes, data on reported symptoms, and data on unplanned healthcare encounters available (figure 1). There were 54 (1.3%) participants who did not die but were lost to follow-up by 60 days and had previous follow-up so were included in our analyses. There were 4105 (96.7%) children who survived and 138 (3.3%) died within 60 days of hospital discharge (median time from hospital discharge to death 30 days, IQR 15, 45 days). Enrolment was approximately equal at each site (47.0% in Tanzania and 53.0% in Liberia). The median age of enrolled participants was 1 month (IQR 0, 9 months) and 43.8% were female (table 1).

### Symptoms following hospital discharge

Participants who died within 60 days of hospital discharge were more likely to ever report difficulty breathing, refusal to eat/drink/breastfeed, diarrhoea, vomiting, abdominal pain, fussiness, seizures, and caregiver-reported weight loss for the child during any follow-up call after discharge than participants who survived (p<0.05 for all comparisons; table 2). Participants who survived 60 days after hospital discharge were more likely to have no reported symptoms at any follow-up call (p<0.001). Similar patterns were observed when comparisons were restricted to those who were discharged as neonates and when restricted to infants and children (table 2) and at each follow-up time point (online supplemental table 1). In multivariable analyses adjusting for site, sociodemographic factors, and clinical factors, for every additional symptom reported following discharge, there was a 35% greater likelihood of post-discharge mortality (adjusted OR [aOR] 1.35, 95% confidence interval [CI] 1.10 to 1.66; p=0.004; table 3).

Participants who died had a longer median proportion of days with cough after discharge (8.9% of days of follow-up, IQR 5%, 16.7% of days of follow-up) than those who survived and had cough (median 5% of days of follow-up, IQR 3.3%, 6.7% of days of follow-up; p<0.001) as well as longer median proportion of days with abdominal pain than those who survived (median 20% of days of follow-up, IQR 11%, 36% of days of follow-up versus median 3% IQR 2%, 5% of days of follow-up, respectively; p=0.01; online supplemental table 2).

**Table 2** Symptoms ever reported during follow up calls at days 7, 14, 30, 45, and 60 among participants

| Symptoms reported by caregiver | Survived 60 days after discharge, n (%) | Died within 60 days after discharge, n (%) | P value* |
|---|---|---|---|
| **All participants** | | | |
| None | 2134 (52) | 23 (17) | <0.001 |
| Common cold | 491 (12) | 15 (11) | 0.70 |
| Cough | 1160 (28) | 41 (30) | 0.71 |
| Difficulty breathing | 85 (2.1) | 50 (36) | <0.001 |
| Fever | 1204 (29) | 48 (35) | 0.17 |
| Refusal to eat, drink or breastfeed | 738 (18) | 49 (36) | <0.001 |
| Pus draining from ear | 25 (0.6) | 1 (0.7) | 0.58 |
| Vomiting | 84 (2.0) | 10 (7.2) | <0.001 |
| Abdominal pain | 24 (0.6) | 8 (5.8) | <0.001 |
| Anaemia | 8 (0.2) | 1 (0.7) | 0.26 |
| Diarrhoea | 35 (0.9) | 4 (2.9) | 0.04 |
| Fussiness | 7 (0.2) | 7 (5.1) | <0.001 |
| Injury | 5 (0.1) | 0 (0) | >0.99 |
| Jaundice | 20 (0.5) | 4 (2.9) | 0.01 |
| Rash | 26 (0.6) | 1 (0.7) | 0.59 |
| Seizure | 17 (0.4) | 5 (3.6) | <0.001 |
| Weight loss | 2 (<0.1) | 2 (1.4) | 0.01 |
| Other | 35 (0.9) | 10 (7.2) | <0.001 |
| **Neonates only** | | | |
| None | 1259 (56) | 13 (18) | <0.001 |
| Common cold | 271 (12) | 10 (14) | 0.62 |
| Cough | 572 (26) | 19 (27) | 0.82 |
| Difficulty breathing | 32 (1.4) | 25 (35) | <0.001 |
| Fever | 605 (27) | 24 (34) | 0.21 |
| Refusal to eat, drink or breastfeed | 362 (16) | 25 (35) | <0.001 |
| Pus draining from ear | 12 (0.5) | 1 (1.4) | 0.33 |
| Vomiting | 31 (1.4) | 4 (5.6) | 0.021 |
| Abdominal pain | 15 (0.7) | 3 (4.2) | 0.016 |
| Anaemia | 4 (0.2) | 0 (0) | >0.99 |
| Diarrhoea | 7 (0.3) | 1 (1.4) | 0.22 |
| Fussiness | 2 (<0.1) | 7 (9.9) | <0.001 |
| Injury | 0 (0) | 0 (0) | – |
| Jaundice | 17 (0.8) | 2 (2.8) | 0.11 |
| Rash | 11 (0.5) | 1 (1.4) | 0.31 |
| Seizure | 0 (0) | 0 (0) | – |
| Weight loss | 1 (<0.1) | 1 (1.4) | 0.061 |
| Other | 12 (0.5) | 3 (4.2) | 0.010 |
| **Infants and children only** | | | |

Continued

**Table 2** Continued

| Symptoms reported by caregiver | Survived 60 days after discharge, n (%) | Died within 60 days after discharge, n (%) | P value* |
|---|---|---|---|
| None | 875 (47) | 10 (15) | <0.001 |
| Common cold | 220 (12) | 5 (7.5) | 0.28 |
| Cough | 588 (32) | 22 (33) | 0.82 |
| Difficulty breathing | 53 (2.8) | 25 (37) | <0.001 |
| Fever | 599 (32) | 24 (36) | 0.52 |
| Refusal to eat, drink or breastfeed | 376 (20) | 24 (36) | 0.002 |
| Pus draining from ear | 13 (0.7) | 0 (0) | >0.99 |
| Vomiting | 53 (2.8) | 6 (9.0) | 0.015 |
| Abdominal pain | 9 (0.5) | 5 (7.5) | <0.001 |
| Anaemia | 4 (0.2) | 1 (1.5) | 0.16 |
| Diarrhoea | 28 (1.5) | 3 (4.5) | 0.090 |
| Fussiness | 5 (0.3) | 0 (0) | >0.99 |
| Injury | 5 (0.3) | 0 (0) | >0.99 |
| Jaundice | 3 (0.2) | 2 (3.0) | 0.011 |
| Rash | 15 (0.8) | 0 (0) | >0.99 |
| Seizure | 17 (0.9) | 5 (7.5) | <0.001 |
| Weight loss | 1 (<0.1) | 1 (1.5) | 0.068 |
| Other | 23 (1.2) | 7 (10) | <0.001 |

*Calculated with $\chi^2$ test.

## Unplanned healthcare encounters following hospital discharge

Among all participants, there were a total of 568 unplanned hospital admissions, 245 unplanned clinic visits, and 2589 pharmacy visits during the 60 days following the index hospital discharge. Among 457 participants with unplanned hospital admissions following discharge, 358 (78.3%) had one hospital admission and one (0.2%) had four unplanned hospital admissions (median one (IQR 1, 1). There were 214 participants who made a total of 245 unplanned clinic visits following discharge (184 [4.3%] had one unplanned clinic visit, 29 [0.7%] had two unplanned clinic visits and one [0.02%] had three unplanned clinic visits).

Participants who died following hospital discharge were more likely to be readmitted (49%, n=68/138 vs 9.5%, n=389/4105; p<0.001), to be readmitted and survive at least one readmission (35%, n=48/138 vs 9.5%, n=389/4105; p<0.001), or to be taken to a clinic (9.4%, 13/168 vs 4.9%, n=201/4105, p=0.02; figure 2A). Conversely, participants who survived were more likely not to seek additional care following hospital discharge (52.0%, n=2134 vs 17.0%, n=23/168, p<0.001). These same patterns were observed in subgroup analyses

**Table 3** Multivariable logistic regression analyses of factors associated with all-cause, 60-day, post-discharge mortality

| | OR (95% CI) | P value | Adjusted OR (95% CI) | P value |
|---|---|---|---|---|
| Site | | | | |
| Liberia | 1.06 (0.75 to 1.49) | 0.75 | 1.64 (0.91 to 2.92) | 0.10 |
| Tanzania | Referent | | Referent | |
| Sociodemographic factors | | | | |
| Sex | | | | |
| Female | Referent | | | |
| Male | 1.09 (0.77 to 1.56) | 0.61 | | |
| Participant age at discharge in months | 0.99 (0.98 to 1.01) | 0.31 | | |
| Caregiver age in years | 1.01 (0.99 to 1.03) | 0.41 | | |
| Caregiver education level | | | | |
| Completed college/university or vocational/technic | Referent | | Referent | |
| Completed/some primary school | 1.49 (0.72 to 3.38) | 0.31 | 1.47 (0.68 to 3.43) | 0.35 |
| Completed/some secondary school | 1.74 (0.91 to 3.77) | 0.12 | 1.66 (0.84 to 3.69) | 0.17 |
| No formal schooling | 2.64 (1.14 to 6.42) | 0.03 | 2.69 (1.08 to 6.96) | 0.03 |
| Other | 1.64 (0.09 to 9.10) | 0.65 | 0.45 (0.01 to 4.89) | 0.58 |
| Clinical factors | | | | |
| Number of discharge diagnoses | | | | |
| 1 only | Referent | | Referent | |
| 2 only | 1.12 (0.73 to 1.73) | 0.60 | 0.84 (0.51 to 1.37) | 0.48 |
| 3 or more | 1.42 (0.94 to 2.17) | 0.10 | 0.79 (0.48 to 1.31) | 0.37 |
| Infectious diagnosis during hospital admission | | | | |
| No | Referent | | | |
| Yes | 0.95 (0.67 to 1.35) | 0.77 | | |
| Disposition from initial hospital admission | | | | |
| Discharge | Referent | | Referent | |
| Against medical advice | 5.49 (3.22 to 8.92) | <0.001 | 8.06 (3.87 to 16.3) | <0.001 |
| Number of reported symptoms after hospital discharge | 1.68 (1.49 to 1.91) | <0.001 | 1.35 (1.10 to 1.66) | 0.004 |
| Healthcare seeking after discharge | | | | |
| Time to nearest health facility in hours | 0.99 (0.88 to 1.11) | 0.83 | | |
| Type of health facility nearest home | | | | |
| Pharmacy | 0.92 (0.53 to 1.72) | 0.79 | | |
| Clinic | 0.78 (0.42 to 1.53) | 0.46 | | |
| Hospital | Referent | | | |
| Other | 0 (0.00 to 1.70) | 0.98 | | |
| Highest type of healthcare sought | | | | |
| Hospital | 2.52 (1.29 to 5.56) | 0.01 | 3.42 (1.55 to 8.47) | 0.004 |
| Clinic | 0.59 (0.22 to 1.58) | 0.29 | 0.34 (0.11 to 1.02) | 0.05 |
| Pharmacy | 0.35 (0.17 to 0.79) | 0.01 | 0.11 (0.05 to 0.30) | <0.001 |
| None but had symptoms | Referent | | Referent | Referent |
| None and had no symptoms | 0.16 (0.07 to 0.36) | <0.001 | 0.22 (0.08 to 0.60) | 0.002 |

among neonates alone and infants and children alone (figure 2B,C).

Participants who were readmitted had 3.42 greater odds (95% CI 1.55 to 8.47; p=0.004) of post-discharge mortality than those who did not seek care and had symptoms (table 3). Participants whose caregivers did not seek care and had no symptoms (aOR 0.22, 95% CI 0.08 to 0.60; p=0.002) had significantly lower odds of post-discharge mortality than those who did not seek additional clinical care but had symptoms. Moreover,

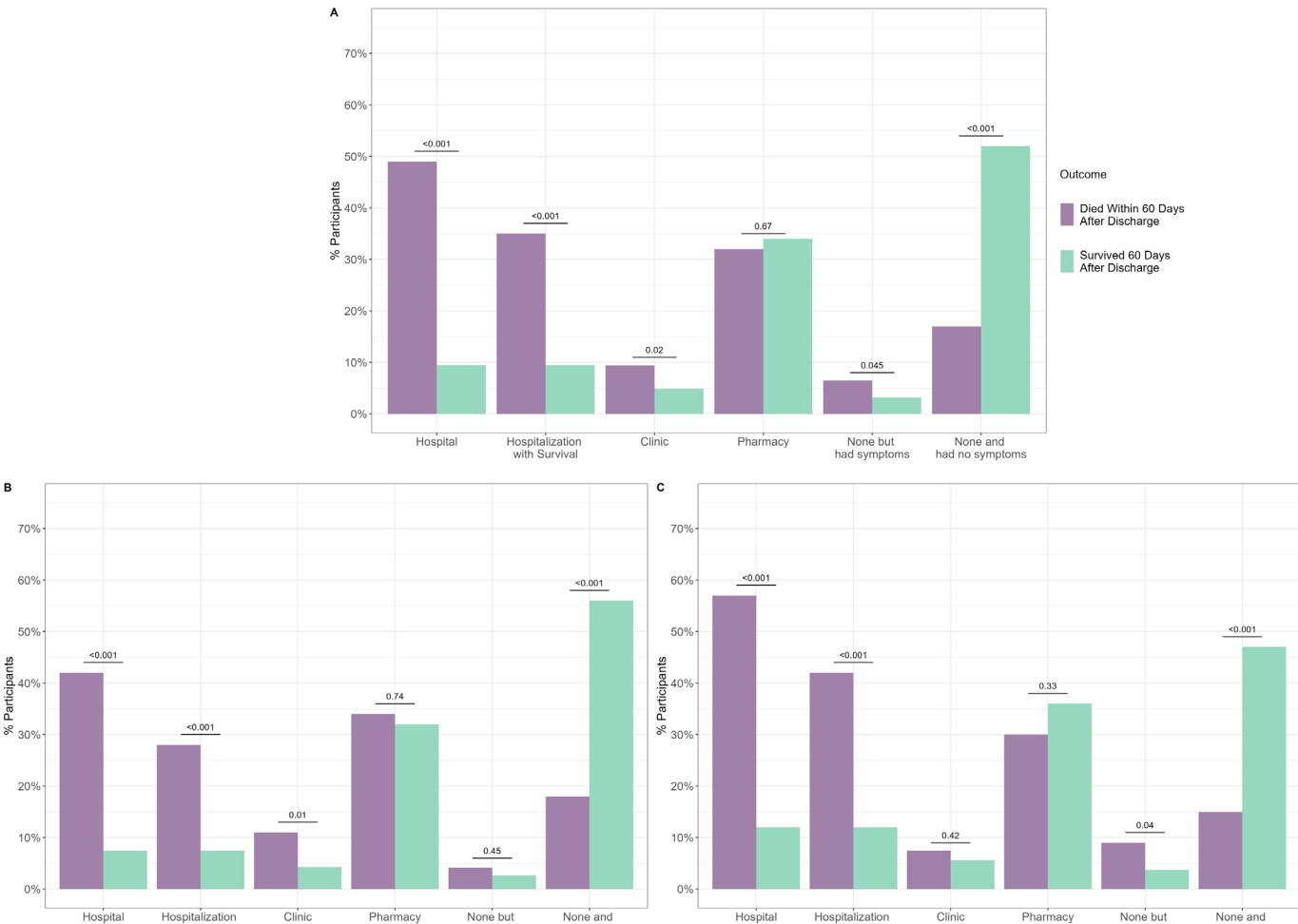

**Figure 2** Comparison of healthcare encounters among children who died and those who survived up to 60 days following hospital discharge among (A) all participants, (B) infants and children only, and (C) neonates only.

**Table 4** Comparison of healthcare encounters among children who died and those who survived up to 60 days following hospital discharge among participants who were discharged against medical advice

| Type of healthcare sought | Survived 60 days after discharge, n (%) | Died within 60 days after discharge, n (%) | P value |
|---|---|---|---|
| Hospital | 0 (0) | 5 (25) | <0.001 |
| Clinic | 12 (9.6) | 6 (30) | 0.02 |
| Pharmacy/herbs | 86 (69) | 13 (65) | 0.73 |
| None but had symptoms | 2 (1.6) | 1 (5.0) | 0.36 |
| None and had no symptoms | 32 (26) | 2 (10) | 0.16 |

participants whose caregivers with no formal education had 2.69 greater odds of post-discharge mortality than those who completed college/university or vocational/technic school (95% CI 1.08 to 6.96; p=0.03).

At each of the follow-up periods, seeking care at a hospital was more common among participants who died and seeking care at a pharmacy was more common among those who survived (online supplemental table 3). There was no significant difference in the survival status of participants when stratified by the amount of time passed since hospital discharge in Kaplan-Meier analyses. A small minority (0.9%, n=38/4243) of participants sought care at each follow-up call, of which 68.4% (n=26/38) died in the 60-day post-discharge period.

## Discharge against medical advice subgroup analysis

Caregivers who took their child home from the hospital against medical advice during the initial hospitalisation had over eight times greater odds of post-discharge mortality (aOR 8.06, 95% CI 3.87 to 16.3; p<0.001). Our subgroup analyses of participants who were discharged against medical advice demonstrated that difficulty breathing and refusal to eat/drink/breastfeed were more common among those participants who died than among those who survived (online supplemental table 4) and that hospital readmissions and unplanned clinic visits were more common among those who died (table 4), similar to the overall population.

## DISCUSSION

In our study of neonates, infants, and young children who were discharged from referral hospitals in Dar es Salaam, Tanzania and Monrovia, Liberia, there was a high burden of symptoms and unplanned healthcare encounters following hospital discharge. Each reported symptom after hospital discharge was associated with a greater likelihood of post-discharge mortality. Moreover, there were clear differences in healthcare seeking locations among children who died following discharge compared with those who survived. Young children whose caregivers left the hospital against medical advice during an index hospitalisation had the greatest odds of post-discharge mortality; however, their symptoms and healthcare seeking patterns following hospital discharge did not differ from those who did not.

Although several studies have described factors present during an initial hospitalisation that confer greater risk of post-discharge mortality among young children in sub-Saharan Africa,[3 4 14] few have compared post-discharge symptoms among young children who died following discharge to symptoms among those who survived.[15] A recent study conducted in Uganda provided a description of the results of verbal autopsy among young children who experienced post-discharge mortality,[16] yet a description of symptoms before the child's death was lacking. Our findings suggest that following discharge, children who had caregiver-reported difficulty breathing had the greatest risk of post-discharge mortality, and other symptoms such as diarrhoea, vomiting, and refusal to eat/drink/breastfeed were also more common among young children who died following discharge than those who survived at least 60 days.

Although our study population included young children admitted for any reason, our results align with those from a cohort study of 369 children aged <5 years followed after hospitalisation for severe pneumonia or malnutrition in Bangladesh that suggested that nearly 75% of children who died following discharge had new onset difficulty breathing, nearly half had vomiting or diarrhoea, and nearly half had poor feeding.[15] However, the frequency of such symptoms among young children who survived was not assessed in that study. Given the differences we observed in symptoms among young children who died following discharge, novel approaches to monitoring for new or persistent symptoms must be explored, which may include follow-up telephone calls, community health volunteer/worker home visits, or the use of telephone applications to screen for difficulty breathing, gastrointestinal symptoms, and difficulty feeding may identify young children at risk for post-discharge mortality. Moreover, as post-discharge mortality was common despite additional healthcare seeking, efforts to improve clinical care quality during the index hospitalisation and the implementation of risk assessment tools to better identify young children at risk for post-discharge mortality are warranted.[6 16]

There were clear differences in healthcare seeking patterns among young children who died following hospital discharge compared with those who survived. Children who died following an initial hospital discharge were more likely to be readmitted to a hospital or to be seen in clinic for an unplanned encounter. This variation in healthcare seeking likely relates to caregiver perception of illness severity, with more severe illness prompting the seeking of a higher level of clinical care. Results from our study suggest that nearly 60% of young children who died following hospital discharge sought additional clinical care at a hospital, which aligns with work conducted in Uganda demonstrating that 72% of young children who died following hospital discharge presented to a hospital for additional clinical care.[17] However, unlike our study, prior studies have not compared healthcare seeking among those who died to those who survived.[18–24] Given the high proportion of children who died during readmission, clinicians should have heightened concern for young children who present to a hospital with 60 days following a recent hospital discharge in such settings.

Social factors, including leaving the hospital against medical advice and no formal education among caregivers, were independently associated with post-discharge mortality among young children enrolled in this study. Although our study was not designed to elucidate why caregivers left the hospital against medical advice, prior work suggests that social reasons such as perceived futility of clinical care or inability to pay medical bills may contribute to the phenomenon of leaving against medical advice.[25] Similar to the results from prior studies including pooled results from a systematic review that assessed the association between caregiver education and all-cause childhood mortality in low- and middle-income countries, we found that low caregiver education was associated with greater risk of childhood mortality.[26 27] Our study adds to the literature as prior studies have not demonstrated an association between caregiver education level specifically following hospital discharge.[3 4]

### Limitations

Although our study provides novel insights into symptomatology and unplanned healthcare encounters among young children who died following hospital discharge, our results should be interpreted in the context of several limitations. As we relied on caregiver report for all symptoms, it is possible that some symptoms may not have been recognised or that some may have been underreported or over-reported. We also did not determine the reasons caregivers did or did not seek additional clinical care following hospital discharge. Prior studies suggest that socioeconomic barriers, perceived suboptimal health services, and negative experiences with healthcare facilities may prevent caregivers from seeking additional clinical care.[28 29] Besides, our study may have a limited

external validity because it may not represent patterns in other resource-limited settings beyond sub-Saharan Africa. Fianlly, our study was conducted at two referral hospitals in Tanzania and Liberia and may not represent patterns of symptoms or unplanned healthcare encounters in rural or other settings in sub-Saharan Africa.

## CONCLUSIONS

Young children in Tanzania and Liberia experienced substantial morbidity following hospital discharge. There were clear differences in symptomatology among young children who died following hospital discharge compared with those who survived, which calls attention to the need for targeted surveillance for persistent or new symptoms to identify young children at risk for post-discharge mortality. Such an approach may present opportunities for intervention, including follow-up telephone calls and community healthcare volunteer/worker home visits, to reduce the burden of post-discharge mortality among young children in sub-Saharan Africa.

**Author affiliations**
[1]Department of Paediatrics and Child Health, Muhimbili University of Health and Allied Sciences, Dar es Salaam, United Republic of Tanzania
[2]Department of Pediatrics, John F. Kennedy Medical Center, Monrovia, Liberia
[3]Accident and Emergency Department, Aga Khan Health Services, Dar es Salaam, United Republic of Tanzania
[4]Department of Emergency Medicine, Muhimbili University of Health and Allied Sciences, Dar es Salaam, United Republic of Tanzania
[5]Departments of Nutrition and Global Health and Population, Harvard T.H. Chan School of Public Health, Boston, Massachusetts, USA
[6]Pediatric Biostatistics Core, Department of Pediatrics, Emory University, Atlanta, Georgia, USA
[7]Division of Emergency Medicine, Boston Children's Hospital, Boston, Massachusetts, USA
[8]Departments of Pediatrics and Emergency Medicine, Harvard Medical School, Boston, Massachusetts, USA
[9]Division of Pediatric Emergency Medicine, Emory University School of Medicine, Atlanta, Georgia, USA
[10]Department of Emergency Medicine, Children's Healthcare of Atlanta, Atlanta, Georgia, USA
[11]Emory Global Health Institute, Emory University, Atlanta, Georgia, USA
[12]Hubert Department of Global Health, Rollins School of Public Health, Emory University, Atlanta, Georgia, USA
[13]Infectious Diseases and Oncology Research Institute, University of the Witwatersrand, Johannesburg, South Africa
[14]Center for Nutrition, Division of Gastroenterology, Hepatology, and Nutrition, Boston Children's Hospital, Boston, Massachusetts, USA

**Acknowledgements** We would like to thank the families and children for their participation in this study. We also thank the study staff who contributed to this study.

**Contributors** RK, RCI, AS, EG, HKM, CRS, MN, CPD, KPM and CAR conceptualised and designed the study. RK, CAR, RCI, JK, AS, EG, CRS, KPM and CPD oversaw data collection and verified the underlying data. CAR and AM verified the underlying data. AM conducted the statistical analyses. RK wrote the first draft of the manuscript. RK, RCI, JK, Y-JGC-N, AS, EG, HKM, CRS, AM, MN, CRM, CGW, RFB, CPD, KPM and CAR interpreted the data, reviewed and provided input to the final draft. RK had final responsibility for the decision to submit for publication. RK is responsible for the overall content as the guarantor and accepts full responsibility for the work, the conduct of the study, and had access to the data, and controlled the decision to publish.

**Funding** The authors wish to acknowledge funding for this work from the National Institutes of Health (K24 DK104676 and P30 DK040561 to CPD and K23HL173694 to CAR), the Boston Children's Hospital Global Health Programme to CAR, the Palfrey Fund for Child Health Advocacy to CAR and the Emory Pediatric Research Alliance Junior Faculty Focused Award to CAR.

**Competing interests** None declared.

**Patient and public involvement** Patients and/or the public were not involved in the design, or conduct, or reporting or dissemination plans of this research.

**Patient consent for publication** Consent obtained from parent(s)/guardian(s).

**Ethics approval** This study involves human participants. The study received ethical clearance from the Tanzania National Institute of Medical Research (#NIMR/HQ/R8a/Vol.IX/3494), the Muhimbili University of Health and Allied Sciences Research and Ethics Committee (#307/323/01), the John F. Kennedy Medical Center Institutional Review Board (#08062019) and the Boston Children's Hospital Institutional Review Board (#P00033242), and the use of deidentified data was exempted from review by the Emory University Institutional Review Board (no number provided for exempted studies). Participants gave informed consent to participate in the study before taking part.

**Provenance and peer review** Not commissioned; externally peer reviewed.

**Data availability statement** Data are available upon reasonable request. Data may be made available upon reasonable request to the corresponding author.

**Author note** This manuscript is an honest and accurate account of the study being reported. No aspects of this study have been omitted or withheld.

**ORCID iD**
Chris A Rees http://orcid.org/0000-0001-6449-0377

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
