## [Reviewer comments · BMJ Paediatrics Open]

ARTICLE DETAILS

TITLE (PROVISIONAL)	Morbidity and Unplanned Healthcare Encounters After Hospital Discharge Among Young Children in Dar es Salaam, Tanzania and Monrovia, Liberia
AUTHORS	Kisenge, Rodrick Ideh, Readon Kamara, Julia Coleman-Nekar, Ye-Jeung Samma, Abraham Godfrey, Evance Manji, Hussein Sudfeld, Christopher Westbrook, Adrianna Niescierenko, Michelle Morris, Claudia Whitney, Cynthia Breiman, Robert Duggan, Christopher Manji, Karim Rees, Chris

VERSION 1 – REVIEW

REVIEWER	Dr. Elizabeth M Keating University of Utah Department of Pediatrics, Division of Pediatric Emergency Medicine 295 Chipeta Way Salt Lake City Utah 84112-9057 United States
REVIEW RETURNED	16-Mar-2024

GENERAL COMMENTS	Overall, this is a well-designed and well written study that will add to the literature. The authors do a fantastic job of selling its novelty and contribution to the literature. I think it should be published after the authors address my minor comments below. Abstract - First sentence: Is this true in HICs, LMICs, or all settings? Also, little attention by whom? The medical community? Public health community?- In the objective sentence, consider adding a bit more detail about when symptoms and healthcare encounters are considered, such as "...encounters on routine follow up phone calls among children..."- Results first sentence, consider adding "were" between participants and enrolled.- Conclusion: surveillance for symptoms by whom? Parents or the
---

	medical system? What this study adds? - The second bullet is confusing as you say that young children who did not seek additional care had greater odds of post-discharge mortality than those who did seek additional care. How are these two groups different? How this study might affect practice? - Again, who would perform the surveillance you suggest? Be more specific with this recommendation. Introduction - First sentence: Is this just in Sub-Saharan Africa, or are there studies in other resource-denied settings? - Line 8: Within how many months did these studies show increased mortality? - Lines 10-12: Have these studies on risk stratification at the time of hospital discharge shown a decrease in mortality? - Overall a clear, succinct, and well written intro section. Methods - Line 40: Consider adding “young children <5 years were enrolled”. - Study setting: can you explain why these two hospitals were selected and not others in the region? Why these two? - Line 21: How close to hospital discharge were children enrolled? - Can you describe the consent process a little bit more? Written or verbal? Who consented patients? - Why was the sample size of 1000 of each age group at each site chosen? Was a power calculation performed? - Line 36: Here is the consent information. What if a caregiver could not write in Tanzania? How was oral consent documented in Liberia? - Could you describe caregivers – were these parents of the children or potential other family members? - Did you have any participants who you discovered on follow-up that they had died, and then the caregiver was unable to answer the questions due to emotional distress? - Was it the same caregiver who answered each phone call? How did you determine this? - For statistical analysis, why were analyses performed in R and SAS? No need to add to manuscript, but I am just curious Results - Were any lost to follow up, such as you could not reach them via cell or home visit? - In Fig 1, what were the reasons for not approached for enrollment? - Table 1: What is the difference between a clinic and a health center in the health facility section? Is a clinic a dispensary? - Table 3: Now the health facility options are different – a clinic and a pharmacy. This is inconsistent with the options in table 1. I suggest making consistent. - Line 18: “One” is spelled out while other smaller numbers are presented numerically. I would be consistent. - Lines 14-22: There are a lot of results presented here and I got a bit lost in the many numbers. I would consider paring down things if possible and presenting the most important results. Perhaps consider presenting in a table? - Line 37: There is an extra word here “...whose caregivers who had no..”
--	---

	 - I'm not familiar with the vocational/technic level of education. Could you explain? - It would be interesting to know when the mortality of the patients who died was. Was it at the first follow up or not until the full 60 days? - I honestly do not like the Sankey diagram. I do not find it helpful in visualizing the results and do not draw any significant conclusions from it beyond what is presented in the text. I would suggest removing it. - The discharge against medical advice subgroup analysis is fascinating! It definitely has clinical implications. Was there approximately the same rates of leaving AMA at the two sites? Any cultural factors that may contribute? A qualitative study on this patient group would be so interesting to explore reasons why. Discussion  - Be careful about restating the results in the first paragraph. Synthesize rather than restating what you have already said above. - Lines 49-55: Since children who sought care in your study had greater mortality, the argument that children at risk should seek care doesn't seem to make total sense to me. Children are seeking care but are still dying. Shouldn't something else be suggested in this case to decrease mortality? Improved care, etc? - Are there any examples of such interventions such as follow up phone calls, community health worker visits, or telephone applications to screen for symptoms? If so, can you add these to your discussion? - Paragraph lines 20-30: Good discussion of the AMA population. However, in lines 27-30 you mention that your study is similar to some studies, and then different to others. You do not describe the suspected reasons for these, which would add to the discussion. Limitations  - Consider also adding that the study may not represent patterns in other resource-denied settings beyond Sub-Saharan Africa (external validity). Conclusions  - Strong conclusions section. To the last sentence you could consider adding a bit about the proposed interventions, such as "...interventions such as follow-up telephone calls, CHW visits, etc".
--	--

REVIEWER	Dr. Huang Chuan-Chin Brigham and Women's Hospital Medicine 75 Francise street Boston Massachusetts 02115-6195 United States
REVIEW RETURNED	22-Mar-2024

GENERAL COMMENTS	General comments: The authors performed a lot of tests and reported the results nicely, but some of the results were speculative and lacked obvious clinical applications. For example, it is intuitive that participants who died
---

	had an increased risk of readmissions compared to those who survived, because most people who were dying were likely to be admitted to the hospital. However, such findings have little clinical implication. Please reconsider all the measurements that were compared between people who died and survived, keeping only the measurements that allow potential interventions and are clinically useful. Page 7 of 28 1. Line 20: How about people with "29 days to aged 1"? Page 9 of 28 2. Lines 3-9: These analyses were subject to Immortal time bias. Participants who died had a shorter follow-up time than those who survived. Therefore, participants who died were likely to have a shorter duration for all measurements due to the immortal time bias. "Total survival days" for each arm as the denominator of all these measurements need to be considered. In other words, the authors need to report the percentage of days with cough, abdominal pain, or difficulty breathing. 3. Lines 24-30: Participants who died were more likely to have a readmission before they died. If the non-survivors died during such readmissions, I am not sure how this result would be useful. If the authors aim to use readmission as a potential indicator to prevent future deaths, the authors may consider repeating the analysis but restrict the readmissions to those in which the participants were discharged successfully. 4. Lines 33-35: A comparison between participants who had symptoms and sought care vs. those who had symptoms but did not seek care seems to be more important and useful. 5. Lines 48-54: Not sure why these comparisons were done. Page 10 of 28 6. Lines 5-13: The results were put in supplementary documents but seem to be more important than the results shown in Page 9 lines 48-54.
--	---

VERSION 1 – AUTHOR RESPONSE

Thank you for pointing this out. In hopes of eventual acceptance and in order to facilitate the article processing charges, Dr. Rees (senior author) submitted the manuscript because he will access the funds needed to pay the article processing charges. Our hope is that Dr. Kisenge will be listed as the corresponding author in the eventual publication. Kindly advise how we may best go about corroborating this.

e appreciate your consideration of our manuscript. Additionally, we appreciate the thoughtful comments made by the Reviewers and we have responded to each of them as outlined below. We are certain that incorporating their feedback has greatly strengthened our manuscript.

Thank you for this comment, we have made changes to address this and now it reads “Researchers and healthcare providers have paid little attention to morbidity and unplanned healthcare encounters for children following hospital discharge in low- and middle-income countries.”

In response to the Reviewer’s comment, we have adjusted the objective sentence, which now reads “Our objective was to compare symptoms and unplanned healthcare encounters among children age <5 years who survived with those who died within 60 days of hospital discharge through follow up phone calls.”

We have added the word “were” to the first sentence of the Results and it now reads, “A total of 4,243 participants were enrolled and had 60-day vital status available; 138 (3.3%) died.”

We have made changes to the Conclusion and it now reads “Surveillance for symptoms and repeated admissions following hospital discharge by healthcare providers is crucial to identify children at risk for post-discharge mortality.”

We have edited the sentence and it now reads “Young children who had symptoms and did not seek additional care following hospital discharge or who were readmitted had greater odds of post-discharge mortality than those who did not have symptoms.”

We have revised the sentence to make it specific and it now reads “Surveillance by healthcare providers following hospital discharge focused on new or persistent symptoms may better identify young children at risk for post-discharge mortality”

We have adjusted the language as requested. The first sentence of the Introduction now reads, “There is emerging evidence that the months following a hospitalization for an illness represent a vulnerable time in the lives of young children in low- and middle-income countries including those in sub-Saharan Africa.”

We appreciate the Reviewer’s comment. Although prior studies have included heterogeneous follow up periods, the majority have assessed this outcome within six months or fewer. To this effect, we have changed the sentence that was on line 8 to read, “As many as 1-18% of young children die within six months of hospital discharge in sub-Saharan Africa.”

The reviewer raises an important question. The cited references have focused on the identification of risk factors for post-discharge mortality and have not assessed the impact of implementing risk assessment tools. The study of post-discharge mortality is largely still in the phase of accurately identifying children at risk. Although there have been studies assessing the efficacy of antimalarial prophylaxis following discharge, this specific population is too narrow to link to the present study.

We appreciate this comment from the Reviewer.

We have made the suggested change and now the line in Study Design reads, “We conducted a planned secondary analysis of a prospective observational cohort study in which neonates, infants, and young children aged <5 years were enrolled at the time of hospital discharge and were followed up to 60 days through telephone calls made to caregivers (i.e., the individual who accompanied the child during the hospital admission and identified as a primary caregiver).”

We have explained why the two hospitals were selected “These two hospitals were selected because of long-standing collaborative relationships among the investigators, their locations being in similar settings, both serving as national referral hospitals, and having similar numbers of monthly discharges from the neonatal and pediatric wards. Additionally, we aimed to include hospitals in diverse settings in sub-Saharan Africa to overcome inherent limitations in single center, or single region, studies.”

We appreciate Reviewers’s comment. In response, we have made additions to the first sentence under Study Population and Inclusion and Exclusion Criteria which now reads, “Young children <5 years were enrolled near the time of hospital discharge (i.e., ≤24 hours before anticipated discharge).”

We have made changes to the description of the consent process under Study Procedures which now reads, “Prior to any data collection, caregivers who were accompanying participants in the hospital were approached by research staff and asked to provide informed written (in Dar es Salaam) or oral consent (in Monrovia) for the collection of sociodemographic data, hospital data, and to follow up calls after hospital discharge. Oral consent was obtained in Monrovia because of cultural preference and low rates of caregiver literacy.”

We have added this sentence under Study Population and Inclusion and Exclusion Criteria to address the Reviewer’s comment, “The sample size was determined based on estimated post-discharge mortality rates of 5% to develop risk assessment tools including at least five variables to identify a) neonates and b) infants and children who were at risk for post-discharge mortality, as described previously.”

To add further description to the caregivers, we have included the following under Study Design, “...through telephone calls made to caregivers (i.e., the individual who accompanied the child during the hospital admission and identified as a primary caregiver).”

No, thankfully we did not have such an occurrence in our study. Given the word count limitations and the absence of such events, we have not made additional changes based on this comment.

We appreciate this comment and have added clarifying language in the third paragraph under Study Procedures as follows, “During these phone calls, research staff first confirmed that the respondent was the caregiver who had consented to enrollment then inquired about common symptoms and asked about any unplanned healthcare

encounters (e.g., clinic visits for illnesses, unplanned hospital admissions, etc.) during each follow up call using a standard case report form.”

Certain figures were easier to generate and customize using R and most of the computations were conducted using SAS.

We have added the following sentence to the first paragraph of the Results in response to the Reviewer’s query, “There were 54 (1.3%) participants who did not die but were lost to follow up by 60 days but had previous follow up so were included in our analyses.”

These were children who left the hospital before they were approached by our research staff. We have adjusted the language in Figure 1 accordingly.

We thank the Reviewer for pointing this out. The language has been clarified in Table 1 to include the frequency of healthcare seeking encounters at pharmacies as originally intended.

This has been clarified and is now consistent in the Tables.

Numbers <10 have been spelled out throughout the Results section.

We appreciate this comment. We have trimmed the language as requested. The first paragraph under Unplanned Healthcare Encounters Following Hospital Discharge now reads, “Among all participants, there were a total of 568 unplanned hospital admissions, 245 unplanned clinic visits, and 2,589 pharmacy visits during the 60 days following the index hospital discharge. Among 457 participants with unplanned hospital admissions following discharge, 358 (78.3%) had one hospital admission and one (0.2%) had four unplanned hospital admissions (median one (IQR 1, 1). There were 214 participants who made a total of 245 unplanned clinic visits following discharge (184 [4.3%] had one unplanned clinic visit, 29 [0.7%] had two unplanned clinic visits, and one [0.02%] had three unplanned clinic visits).”

We have removed the extra word and now the sentence reads, “Moreover, participants whose caregivers had no formal schooling had 2.69 greater odds of post-discharge mortality than those who completed college/university or vocational/technic (95% CI 1.08, 6.96; P=0.03)”.

We have added further clarification under Table 1 as a footnote as follows, “*Vocational/technic denotes training for specific jobs (e.g., electrician).”

We have added the following to paragraph one in the Results, “There were 4,105 (96.7%) children who survived and 138 (3.3%) died within 60 days of hospital discharge (median time from hospital discharge to death 30 days, interquartile range [IQR] 15, 45 days).”

As suggested, we have removed the Sankey diagram and the corresponding text that described that Figure.

We appreciate the Reviewer’s remarks. We have included the following language to acknowledge this need in the Discussion as follows, “Although our study was not designed to elucidate why caregivers left the hospital against medical advice, prior work suggests that social reasons such as perceived futility of clinical care or inability to pay medical bills may contribute to the phenomenon of leaving against medical advice.”

In response, we have made changes to the first paragraph and it now reads, “In our study of neonates, infants, and young children who were discharged from referral hospitals in Dar es Salaam, Tanzania and Monrovia, Liberia, there was a high burden of symptoms and unplanned healthcare encounters following hospital discharge. Each reported symptom after hospital discharge was associated with a greater likelihood of post-discharge mortality. Moreover, there were clear differences in healthcare seeking locations among children who died following discharge compared to those who survived. Young children whose caregivers left the hospital against medical advice during an index hospitalization had the greatest odds of post-discharge mortality; however, their symptoms and healthcare seeking patterns following hospital discharge did not differ from those who did not”.

We agree with the Reviewer and have moderated this language in paragraph three of the Discussion as follows, “Given the differences we observed in symptoms among young children who died following discharge, novel approaches monitoring must be explored, which may include follow-up telephone calls, community health worker visits, or the use of telephone applications to screen for difficulty breathing, gastrointestinal symptoms, and difficulty feeding may identify young children at risk for post-discharge mortality. Moreover, as post-discharge mortality was common despite additional healthcare seeking, efforts to improve clinical care quality during the index hospitalization and the implementation of risk assessment tools to better identify young children at risk for post-discharge mortality are warranted.”

These interventions have not been implemented previously, to our knowledge. To add clarity to this important point, we have changed the language in paragraph three of the Discussion as follows, “Given the differences we

observed in symptoms among young children who died following discharge, novel approaches monitoring must be explored, which may include follow-up telephone calls, community health worker visits, or the use of telephone applications to screen for difficulty breathing, gastrointestinal symptoms, and difficulty feeding may identify young children at risk for post-discharge mortality.”

We have changed the language in the final two sentences of paragraph five of the Discussion as follows, “Similar to the results from prior studies included pooled results from a systematic review that primarily assessed the association between caregiver education and all-cause childhood mortality in low- and middle-income countries, we found that low caregiver education was associated with increased childhood mortality. Our study adds to the literature as prior studies have not demonstrated an association between caregiver education level specifically following hospital discharge.”

We have made this addition to the limitation “Besides, our study may have a limited external validity because it may not represent patterns in other resource-limited settings beyond sub-Saharan Africa”.

We appreciate this comment from the Reviewer. We have made this addition in the last sentence of the conclusion and it now reads “Such an approach may present opportunities for intervention, like follow-up telephone calls and community health care worker visits, to reduce the burden of post-discharge mortality among young children in sub-Saharan Africa.”

REVIEWER 2 (Dr. Huang Chuan-Chin, Brigham and Women's Hospital)

We appreciate this comment from the Reviewer. We believe that the responses below have added further clarification with a focus on measurements that may be clinically useful.

We appreciate the Reviewer’s comment. We have changed the language under Study Population and Inclusion and Exclusion Criteria to the following, “Participants aged 0 to 28 days at the time of discharge were considered neonates and those 29 days to 59 months of age were considered infants and children in this study.”

We appreciate the Reviewer’s comment. We have re-analyzed our data using the median proportion of days with symptoms as suggested. We have replaced Supplemental Table 2 with these new results and have updated the text in the second paragraph under Symptoms Following Hospital Discharge in the Results as follows, “Participants who died had a longer median proportion of days with cough after discharge (8.9 days, IQR 5, 16.7 days) than those who survived and had a cough (median 5 days, IQR 3.3, 6.7 days; $P<0.001$) as well as longer median proportion of days with abdominal pain (median 20 days, IQR 11, 36 vs. median 3 days IQR 2, 5; $P=0.01$; Supplemental Table 2).”

In response to the Reviewer’s comment, we conducted additional analyses that included a subanalysis of participants who were readmitted and survived at least one readmission. We have included an additional section in Figure 2 to include these analyses. Moreover, we have added the following to the Results in paragraph two under Unplanned Healthcare Encounters Following Hospital Discharge, “Participants who died following hospital discharge were more likely to be readmitted (49%, $n=68/138$ vs. 9.5%, $n=389/4,105$; $P<0.001$), to be readmitted and survive at least one readmission (35%, $n=48/138$ vs. 9.5%, $n=389/4,105$; $P<0.001$), or to be taken to a clinic (9.4%, $13/168$ vs. 4.9%, $n=201/4,105$, $P=0.02$; Figure 2A).”

We have changed the referent in Table 3 as suggested and updated the text with the new adjusted odds ratios.

We have removed the Sankey diagram as suggested and have removed the Results in question.

We appreciate the Reviewer’s recommendation. We have moved what was previously Supplemental Table 5 to the main text (now Table 4).

VERSION 2 – REVIEW

REVIEWER	Dr. Elizabeth M Keating University of Utah Department of Pediatrics, Division of Pediatric Emergency Medicine 295 Chipeta Way Salt Lake City Utah 84112-9057 United States
REVIEW RETURNED	16-May-2024

GENERAL COMMENTS	Thank you to the authors for addressing my comments thoroughly. I am now in agreement with publishing this article.
---

REVIEWER	Dr. Huang Chuan-Chin Brigham and Women's Hospital Medicine 75 Francise street Boston Massachusetts 02115-6195 United States
REVIEW RETURNED	06-May-2024

GENERAL COMMENTS	1. I am not sure that I understand the response to my comment 3. The authors stated "Participants who died had a longer median proportion of days with cough after discharge (8.9 days, IQR 5, 16.7 days) than those who survived and had a cough (median 5 days, IQR 3.3, 6.7 days; $P < 0.001$) as well as longer median proportion of days with abdominal pain (median 20 days, IQR 11, 36 vs. median 3 days IQR 2, 5; $P = 0.01$; Supplemental Table 2)" Median proportion of days should still be a proportion, but not days.
---

VERSION 2 – AUTHOR RESPONSE

Editor(s)' Comments to Author (if any):

A minor issue to be addressed as per the reviewer. This is a really important study and we look forward to getting an updated submission.

Author response: We thank the Editor for their support of our work and we look forward to the publication of this important piece.

Associate Editor

Minor revision needed to address reviewer's query

Author response: We appreciate the Associate Editor's time in this process. We have responded to the Reviewer's query below.

Reviewer 2: Dr. Huang Chuan-Chin, Brigham and Women's Hospital

1. I am not sure that I understand the response to my comment 3. The authors stated "Participants who died had a longer median proportion of days with cough after discharge (8.9 days, IQR 5, 16.7 days) than those who survived and had a cough (median 5 days, IQR 3.3, 6.7 days; P<0.001) as well as longer median proportion of days with abdominal pain (median 20 days, IQR 11, 36 vs. median 3 days IQR 2, 5; P=0.01; Supplemental Table 2)"

Median proportion of days should still be a proportion, but not days.

Author response: We thank Dr. Huang Chuan-Chin for their time spent re-reviewing our manuscript. The Reviewer is correct. The median proportion of days should be a proportion. We have fixed this language in the Results in the second paragraph under Symptoms Following Hospital Discharge as follows:

"Participants who died had a longer median proportion of days with cough after discharge (8.9% of days of follow up, IQR 5%, 16.7% of days of follow up) than those who survived and had cough (median 5% of days of follow up, IQR 3.3%, 6.7% of days of follow up; P<0.001) as well as longer median proportion of days with abdominal pain than those who survived (median 20% of days of follow up, IQR 11%, 36% of days of follow up vs. median 3% IQR 2%, 5% of days of follow up, respectively; P=0.01; Supplemental Table 2)."

We have also made the column headings clearer in Supplemental Table 2 according to the Reviewer's comment.

Reviewer 1: Dr. Elizabeth Keating, University of Utah, University of Utah

Thank you to the authors for addressing my comments thoroughly. I am now in agreement with publishing this article.

Author response: We thank Dr. Keating for their time spent re-reviewing our manuscript.

VERSION 3 – REVIEW

REVIEWER	Dr. Huang Chuan-Chin Brigham and Women's Hospital Medicine 75 Francise street Boston Massachusetts 02115-6195 United States
REVIEW RETURNED	30-May-2024
GENERAL COMMENTS	No further comments.

VERSION 3 – AUTHOR RESPONSE

None